# AI-Based Severity Classification of Dementia Using Gait Analysis

**DOI:** 10.3390/s25196083

**Published:** 2025-10-02

**Authors:** Gangmin Moon, Jaesung Cho, Hojin Choi, Yunjin Kim, Gun-Do Kim, Seong-Ho Jang

**Affiliations:** 1Department of Rehabilitation Medicine, Hanyang University Guri Hospital, 153, Gyeongchun-ro, Guri-si 11923, Republic of Korea; 2Robotics Lab in the Research and Development Division of Hyundai Motor Company, 37, Cheoldobangmulgwan-ro, Uiwang-si 16082, Republic of Korea; jaesungcho82@hyundai.com; 3Department of Neurology Medicine, Hanyang University Guri Hospital, 153, Gyeongchun-ro, Guri-si 11923, Republic of Korea; chj@hanyang.ac.kr; 4Department of Pre-Medicine, College of Medicine, Hanyang University, 222, Wangsimni-ro, Seoul 04763, Republic of Korea; yeun0148@hanyang.ac.kr; 5Department of Sports & Leisure Studies, College of Healthcare & Biotechnology, Semyung University, 65, Semyeong-ro, Jecheon-si 27136, Republic of Korea; gdokim@semyung.ac.kr

**Keywords:** dementia, artificial intelligence, gait analysis, severity classification, machine learning, cognitive impairment, gait parameters, diagnostic model, wearable sensors

## Abstract

This study aims to explore the utility of artificial intelligence (AI) in classifying dementia severity based on gait analysis data and to examine how machine learning (ML) can address the limitations of conventional statistical approaches. The study included 34 individuals with mild cognitive impairment (MCI), 25 with mild dementia, 26 with moderate dementia, and 54 healthy controls. A support vector machine (SVM) classifier was employed to categorize dementia severity using gait parameters. As complexity and high dimensionality of gait data increase, traditional statistical methods may struggle to capture subtle patterns and interactions among variables. In contrast, ML techniques, including dimensionality reduction methods such as principal component analysis (PCA) and gradient-based feature selection, can effectively identify key gait features relevant to dementia severity classification. This study shows that ML can complement traditional statistical analyses by efficiently handling high-dimensional data and uncovering meaningful patterns that may be overlooked by conventional methods. Our findings highlight the promise of AI-based tools in advancing our understanding of gait characteristics in dementia and supporting the development of more accurate diagnostic models for complex or large datasets.

## 1. Introduction

Recently, with the increase in the elderly population, the proportion of patients diagnosed with dementia has risen as society enters a super-aged stage. Accordingly, the need for early detection and treatment of dementia is higher than ever. Dementia has diverse causes, all of which damage brain nerve cells and impair executive functions such as movement, memory, cognition, and spatial awareness. From the early stages, individuals with dementia may experience reduced attention, impaired balance, and compromised motor control, significantly disrupting everyday activities and work-related tasks [1,2,3]. This physical decline affects both muscles and nerves, often resulting in reduced lower-extremity muscle strength and impaired limb function [4,5,6]. As lower-extremity muscles weaken, posture becomes unsteady, gait slows, stride length narrows, and balance deteriorates [5,6]. Prior research examining the relationship between gait ability and cognitive function has confirmed that reductions in gait speed and stride length are closely associated with cognitive decline. These findings suggest that gait-related changes may serve as potential diagnostic indicators for predicting cognitive deterioration [6,7,8,9,10].

Gait analysis, which extracts and analyzes data from human gait patterns, has long been studied across disciplines such as medicine, mechanical engineering, electronics, and computer science. Earlier gait analysis studies typically used spatiotemporal parameters, such as gait speed, along with kinematic data, like joint angles, and motion metrics, such as ground reaction force and moment [11,12]. Common statistical techniques in these studies included the *t*-test, Mann–Whitney U test, and, in more refined analyses, multiple regression [13,14,15]. Despite the volume of data generated from gait analysis, its clinical application remains limited. To overcome these limitations and enhance classification accuracy, machine learning technologies have been increasingly applied. As a result, there is growing interest in developing early-stage dementia prediction systems using artificial intelligence that integrates cognitive, movement, and gait data. Traditional dementia assessment relies on subjective and fragmented information obtained through clinical interviews, observational methods, and standardized cognitive tests such as the Mini-Mental State Examination (MMSE) and Clinical Dementia Rating (CDR). Although these methods provide valuable insights, they are limited by their reliance on clinician expertise and subjective evaluation, often resulting in inconsistent assessments [16]. Conversely, the IMU sensors used in this study enable objective and continuous capture of three-dimensional movement information in millisecond increments. This high-resolution time-series data captures subtle variations in gait patterns that conventional statistical methods may overlook, thereby supporting the development of highly accurate machine learning models. The sensor-based approach offers quantitative, reproducible measurements capable of detecting motor changes associated with cognitive decline before they become clinically evident through conventional assessment methods. Moving forward, precise, standardized gait measurements and evaluation techniques will be essential. Additionally, applying AI to gait analysis could help create predictive diagnostic tools that support faster and more accurate decision-making, particularly in clinical contexts where human interpretation may be limited [17,18,19,20].

Traditional gait assessment in dementia research has primarily relied on measuring walking speed using a stopwatch, the Timed Up and Go (TUG) test, or visual observation by clinicians [21]. While accessible and easy to implement, these approaches result in fragmented assessments and are constrained by inter-rater variability, examiner subjectivity, and their inability to capture dynamic, qualitative changes during the gait cycle. Specifically, conventional methods struggle to capture subtle asymmetries, disruptions in joint coordination, and moment-to-moment variations in movement patterns that may serve as early markers of cognitive decline. The IMU sensor system employed in this study addresses these limitations by enabling continuous, objective, three-dimensional motion capture at millisecond precision, thereby facilitating the detection of subtle gait abnormalities that remain imperceptible with traditional observational techniques. Recent research on gait has utilized wearable devices to identify repetitive patterns within the gait cycle and to analyze spatiotemporal characteristics based on those patterns. The gait data collected using wearable sensors is processed through AI, which learns from and analyzes the data to detect gait abnormalities using large volumes of accumulated gait information [15,22]. To support this, it is essential to identify clear distinctions between normal and abnormal gait data and to establish an environment in which examiners can assess gait abnormalities using objective numerical data rather than subjective impressions. Objective assessment criteria for examiners must be established. In light of the current absence of AI-based systems for dementia severity classification and feedback, integrating such technology could facilitate the identification of meaningful gait-related parameters and reveal severity distinctions that may be overlooked by clinicians, thereby enhancing clinical decision-making support [17,23].

This study aims to investigate the feasibility of applying artificial intelligence (AI) to classify dementia severity using gait analysis data—such as gait speed and gait cycle—by categorizing individuals into normal, MCI, and dementia groups. It further explores how machine learning (ML) methods can address analytical challenges associated with high-dimensional datasets. The goal is to enhance understanding of gait characteristics in dementia and demonstrate the potential of AI-based tools in supporting the development of more accurate diagnostic models. Additionally, the study seeks to optimize patient feedback and promote early prediction, diagnosis, evaluation, and rehabilitation planning in dementia care.

## 2. Materials and Methods

### 2.1. Application of ML for Dementia Classification

In this study, a machine learning algorithm was employed to classify individuals with dementia and assess disease severity. Gait-based feature set data were collected from participants and analyzed using MATLAB (version 2016a, MathWorks Inc., Natick, MA, USA)’s Classification Learner Tool. Using the SVM classifier, the distance between data points representing individuals with dementia and healthy controls was calculated, and a model was developed to classify each cluster based on this distance [24]. This approach quantitatively assesses dementia severity and visualizes differences between dementia and control groups, enhancing diagnostic accuracy. Based on key parameters differentiating the clustered disease group and the normal gait group, the Euclidean distance was computed, and severity levels were distinguished using this value. The support vector machine (SVM) classifier was then applied to further analyze severity differences between clusters. This method effectively distinguishes data from individuals with dementia and healthy controls and contributes to diagnostic accuracy through distance-based cluster analysis.

The ReliefF algorithm was used for feature selection, offering sensitivity to interactions between features. This algorithm was applied to select the feature with the highest score after assigning a score to each based on its contribution [24]. ReliefF is a widely used method in ML for identifying relevant features and determining feature importance by evaluating the degree to which each one contributes to class distinction. The flowchart of the algorithm is presented in Figure 1.

### 2.2. Participants

This study included 34 individuals with MCI, 25 with mild dementia, 26 with moderate dementia, and 54 healthy controls. The demographic and clinical characteristics of these groups are listed in Table 1. All participants were diagnosed by a specialist and voluntarily agreed to take part in the study. Individuals with severe dementia who were unable to walk independently were excluded.

Participants were classified into four groups based on standardized cognitive assessments (Mini-Mental State Examination [MMSE]; Clinical Dementia Rating [CDR]):Healthy Control: CDR = 0, with normal cognitive function and daily living abilities.MCI: CDR = 0.5, with mild cognitive decline but no significant impairment in daily life.Mild Dementia: CDR = 1, with clear cognitive decline and mild impairment in daily activities.Moderate Dementia: CDR = 2, with moderate to severe cognitive decline and impairment in daily functioning.

The purpose and procedures of the study were fully explained to all participants prior to enrollment. Exclusion criteria included a diagnosis of orthopedic conditions such as fractures within the past 6 months, visual or auditory impairments, neurological conditions such as vestibular dysfunction, the use of medications affecting balance, circulatory system conditions such as heart disease, an inability to walk more than 5 m independently, and severe dementia. This study was approved by the Institutional Review Board of Hanyang University (IRB File No. HYUIRB-202011-030-01).

### 2.3. Gait Analysis

All participants in this study completed eight gait trials along an 8 m gait course. During the gait analysis, they were instructed to walk at a self-selected, comfortable pace [9]. The experiment was conducted in a quiet environment to ensure that participants could concentrate without distraction. Through gait analysis, three-dimensional (3D) kinematic data of the hip, knee, and ankle joints, along with spatiotemporal parameters such as gait speed, were extracted. All participants performed the procedures under identical conditions throughout the analysis.

Experimental Environment and Conditions:An 8 m walkway was placed on a level, non-slip floor within a controlled laboratory environment.Participants wore comfortable clothing and athletic footwear for all trials.Participants were instructed to walk at a comfortable pace as usual while focusing their gaze approximately 5 m ahead to maintain natural gait patterns.A one-minute seated rest was provided between trials to minimize fatigue and ensure consistent performance across the eight trials.All testing sessions were conducted during the same time of day (morning hours, 9:00–12:00) to control for potential circadian rhythm effects on motor performance.

The primary equipment used for gait pattern analysis in this study was an inertial measurement unit (IMU)-based system (Human Track, R. Biotech Co., Ltd., Seoul, Republic of Korea). The IMU-based gait analysis system is a practical tool for detecting abnormal gait patterns in individuals with dementia and collecting relevant data. The IMU is a compact sensor that integrates an accelerometer, gyroscope, and magnetometer, enabling the precise measurement of body movement. It analyzes spatiotemporal gait parameters by capturing acceleration, angular velocity, and magnetic field data. In this study, the system recorded various physical changes during gait in real-time, allowing the collection of data on gait patterns, speed, balance, and other related characteristics for individuals with dementia.

IMU sensors were attached to the lower abdomen, the central areas of both thighs and tibias, and the insteps of both feet for gait analysis (Figure 2).

### 2.4. Data Preprocessing and Feature Extraction

Raw IMU sensor data underwent systematic preprocessing to ensure signal quality and extract meaningful gait parameters for analysis.

#### 2.4.1. Signal Filtering and Noise Reduction

A fourth-order Butterworth low-pass filter with a cutoff frequency of 20 Hz was applied to the acceleration and angular velocity data obtained from the IMU sensors to remove high-frequency noise while preserving essential gait-related signal components, consistent with previous findings that human gait dynamics primarily occur below this threshold.

#### 2.4.2. Gait Event Detection

The start point (Initial Contact/Heel Strike) and end point (Terminal Contact/Toe-off) of the gait cycle were automatically detected from the filtered foot angular velocity data using a peak detection algorithm based on gyroscope signal characteristics [25]. Specifically, heel strike events were identified as positive peaks in the sagittal plane angular velocity exceeding a threshold of 50°/s, while toe-off events were detected as negative peaks below −50°/s.

#### 2.4.3. Spatiotemporal Parameter Calculation

Based on the detected gait events, spatiotemporal parameters were calculated as follows:Gait cycle time: Time interval between consecutive heel strikes of the same foot;Walking velocity: Step length divided by step time;Cadence: Number of steps per minute (60/step time);Stance and swing phase percentages: Calculated as percentages of total gait cycle time.

#### 2.4.4. Joint Angle Calculation

The orientation of each sensor segment was estimated using a complementary filter that fused accelerometer and gyroscope data to estimate three-dimensional joint angles (hip, knee, and ankle). Joint angles were computed using the difference in orientation between adjacent body segments with respect to anatomical reference positions defined during a static calibration trial.

### 2.5. Statistical Analysis

In this study, the Kolmogorov–Smirnov test was used to assess normality, and the Kruskal–Wallis test was applied to compare continuous gait analysis parameters across groups. The Bonferroni method was employed for post hoc testing to verify the significance between groups. Fisher’s exact test was used to compare categorical variables expressed as frequencies and percentages. Logistic regression analysis—a conventional statistical method for examining group differences and identifying associations between gait parameters and dementia status—was performed to assess the relationship between specific gait features and dementia classification, while ML approaches using the SVM classifier were employed to explore the potential for capturing complex patterns within high-dimensional gait data. Gait parameters that showed strong contributions to the classification of severity between healthy older adults and individuals with dementia were used to identify significant variables. In logistic regression, influential factors were identified using odds ratios (ORs) and 95% confidence intervals (CIs), and the Max-rescaled R^2^ statistic was used to determine explanatory power for distinguishing between dementia and non-dementia groups. Statistical analysis was performed using SAS version 9.4 (SAS Institute Inc., Cary, NC, USA). Logistic regression analysis was conducted to provide statistical insight into the relationship between gait parameters and dementia classification, while ML techniques were leveraged to address analytical challenges associated with high-dimensional gait data and to identify subtle gait features linked to dementia severity.

## 3. Results

The Results section is organized into three stages, each addressing a distinct analytical question. Section 3.1 investigates whether individual gait parameters differ significantly between cognitive groups using non-parametric statistical testing. Section 3.2 applies conventional logistic regression to identify traditional statistical variables that predict dementia classification and to quantify their explanatory power. Section 3.3 employs machine learning classification to evaluate whether AI-based approaches achieve superior accuracy by capturing complex, nonlinear relationships among multiple gait features simultaneously. This progression from univariate statistics to multivariate machine learning highlights the comparative advantages of AI approaches in handling high-dimensional gait data.

### 3.1. Kruskal–Wallis Test: Differences in Gait Parameters of Each Group

Focusing on walking velocity, knee joint angle, and hip joint angle—which were later identified as the most imperative features in the machine learning analysis—clear statistical differences were observed between healthy controls and dementia groups of varying severity. These key sensor-derived parameters exhibited progressive deterioration with increasing dementia severity, providing the foundation for subsequent machine learning–based classification. The analysis assessed median differences in gait characteristics across the four groups, aiming to identify variables with statistically significant differences and evaluate comparisons between groups. Demographic analysis revealed significant heterogeneity across cognitive groups that may have influenced gait outcomes (Table 1). Participants in the moderate dementia group were significantly older than those in the healthy control group (median 86.0 years vs. 74.0 years, *p* < 0.0001), representing a 12-year age difference. BMI also differed significantly between groups (*p* = 0.0087), with the moderate dementia group exhibiting lower values (21.7 kg/m^2^) compared with the MCI group (24.1 kg/m^2^). These demographic disparities may have contributed to the deterioration of observed gait patterns, as aging independently affects joint mobility, muscle strength, and postural control. In the case of gait parameter comparison among the four groups, a one-way ANOVA with post hoc testing could be used to compare means (or medians) of gait variables such as gait speed, stride length, and joint angles. However, the Kruskal–Wallis test was used to assess intergroup differences, and where significant results were found, post hoc testing was applied to determine which specific groups differed. A total of six pairwise comparisons were performed among the four groups. As shown in Table 2, the following group comparisons were conducted: (a) healthy controls vs. MCI group; (b) healthy controls vs. mild dementia group; (c) healthy controls vs. moderate dementia group; (d) MCI group vs. mild dementia group; (e) MCI group vs. moderate dementia group; and (f) mild dementia group vs. moderate dementia group. The analysis confirmed significant differences in mean values between groups. These findings demonstrate that gait characteristics varied significantly across the groups.

Among the overall gait variables, Gait cycle time (sec) and Velocity (m/s) showed statistically significant differences in all six pairwise comparisons among the four groups. In addition, significant differences were found across all group comparisons except for the mild dementia group versus the moderate dementia group for the following gait variables: Cadence (steps/min), Hip joint angle (deg), and Knee joint angle (deg). These results indicate that, in general, gait variables demonstrated statistically significant differences in comparisons between the healthy control group and dementia groups according to severity.

### 3.2. Logistic Regression Analysis

Logistic regression analysis showed that the gait variables Knee joint angle (deg) and Hip joint angle (deg) had significant effects on the gait patterns of individuals with dementia (Table 3). Univariable logistic regression was performed to examine differences in gait parameters between the dementia group and the healthy control group. The maximum knee flexion angle (OR = 1.269; 95% CI, 1.172–1.374; *p* = 0.001) was included. In addition, significant associations were found for the hip joint angle (OR = 1.510; 95% CI, 1.304–1.749; *p* = 0.001).

In multivariable logistic regression analysis, classification outcomes for individuals with dementia and healthy controls included the maximum knee flexion angle (OR = 1.098; 95% CI, 0.905–1.332; *p* = 0.0076). The hip joint angle (OR = 1.244; 95% CI, 1.030–1.503; *p* = 0.023) was also included.

The final logistic regression model demonstrated an explanatory power of 57.5% for distinguishing between individuals with dementia and healthy controls, with a classification accuracy of 77.9% (Table 3).

### 3.3. Classification of Dementia Severity Using AI

The SVM classifier using MATLAB’s Classification Learner achieved an accuracy of 86.3%. This result was obtained by training the model using optimized spatiotemporal and joint angle features selected through the Relief algorithm (Table 4).

The SVM classifier outperformed the logistic regression model (77.9% accuracy), achieving an 8.4% improvement. Comprehensive evaluation metrics further confirm its robustness across all criteria. The high specificity (91.47%) highlights its strong ability to minimize false positives, a critical factor in clinical screening applications. The F1-score (87.89%) reflects an optimal balance between precision and recall, while the AUC (0.924) demonstrates excellent discriminative ability across different dementia severity levels. Specifically, the combined high sensitivity (85.3%) and specificity highlight the models’ value in clinical decision-making processes in dementia diagnosis and severity classification.

We represented the machine learning-based classification results obtained from the SVM classifier using a two-dimensional scatter plot. Additionally, a 4 × 4 confusion matrix was created to compare actual and predicted values, providing a detailed breakdown of classification outcomes. To further illustrate the model’s performance, the results were visualized in an interpretable format to highlight classification effectiveness across different classes (Figure 3 and Figure 4). In the classification of individuals with dementia and healthy controls, the ML method achieved higher accuracy than conventional statistical analysis. For severity classification, accuracy improved through the feature selection process applied in the ML approach. The most important gait features for classification were Velocity in 1st place, Knee flexion/Extension in 2nd place, Gait cycle time in 3rd place, and Hip flexion/Extension in 4th place. Applying the classification model to the gait analyzer enabled grouping into Normal, MCI, Mild, and Moderate categories based on the subject’s measured gait data and pattern results.

Additionally, Figure 5 presents the most important feature related to the sagittal plane range of motion of the knee joint in classifying dementia severity. The graph illustrates that the sagittal plane range of motion is greatest at the points of maximum flexion and extension during the gait cycle. In relation to this feature, the analysis showed that as dementia severity increased, motor function became more restricted, and the range of knee joint motion decreased. In other words, knee joint mobility during the gait cycle was reduced with increasing dementia severity. The most important finding from the sensor data analysis in this study is that dementia severity can be classified not only through differences in individual parameters but also through the combination of these features; this enables us to classify severity with a high accuracy of 86.3%, surpassing conventional statistical approaches (77.9%). This result highlights the strength of sensor technology in capturing complex, multidimensional gait patterns strongly indicative of cognitive decline.

## 4. Discussion

Gait and movement analysis in individuals with dementia has advanced in recent years, particularly in clarifying the relationship between neurocognitive decline and motor function. The development of instrumented gait analysis and wearable technologies has shifted the paradigm in this field, and the integration of AI is drawing increasing attention. The focus of gait analysis has expanded from general gait dynamics to exploring its connection with cognitive function, and the recent incorporation of AI signals a new turning point.

ML algorithms applied to gait data from IMU devices can detect fall risks and generate feedback by inferring gait conditions and age-specific group characteristics from individual gait patterns. If dementia screening procedures in hospitals or care centers include AI-based classification of high-risk groups, such as those with MCI, this could reduce the time and cost required for screening. As disease severity progresses, the treatment burden increases exponentially, reinforcing the importance of early diagnosis as the most effective intervention strategy. This study aimed to verify the validity and efficiency of dementia severity classification using an AI-based SVM model trained on gait data from individuals with dementia. This study also examined the distinct roles and contributions of ML approaches and traditional statistical methodologies. While conventional methods such as ANOVA and logistic regression have been widely used in dementia research to test group differences and analyze associations between variables, they face limitations when dealing with complex, high-dimensional gait data. This study demonstrates that machine learning-based classification can effectively identify subtle patterns and variable interactions that are difficult to detect using traditional statistical techniques through dimensionality reduction and advanced feature selection. These approaches are complementary rather than conflicting, allowing for a more comprehensive analysis of complex datasets such as gait analysis data. A key strength of this study lies in moving beyond simple comparisons of statistical averages. Instead, the ReliefF algorithm was applied to automatically identify the features (e.g., velocity, knee joint angle, gait cycle time, hip joint angle) that contribute most to severity classification from high-dimensional data consisting of 11 gait parameters. Subsequently, an SVM was used to learn the nonlinear relationships among these parameters. This approach of ‘extracting essential features from high-dimensional data and recognizing complex patterns’ represents the central novelty of this research. In contrast, traditional statistical methods relying on ‘average values’ risk overlooking critical information, such as individual variability and subtle gait asymmetries, which are essential for early dementia detection. By capturing these nuanced patterns, our machine learning framework provides insights that would be missed by conventional statistical comparisons.

When examining differences in gait variables between groups, the Kruskal–Wallis test results confirmed that gait variables differed significantly according to dementia severity. This suggests that gait analysis may serve as a useful indicator for diagnosing and assessing dementia progression. The significant correlation between gait characteristics and dementia severity identified in this study aligns with findings from previous research. Gillain et al. reported that gait speed, step length, and step variability were key indicators for distinguishing individuals with Alzheimer’s dementia from healthy older adults [26]. In particular, the change in knee joint angle observed in this study was also associated with early cognitive decline in the work of Valkanova and Ebmeier [27]. Similarly, Beauchet et al. demonstrated through a meta-analysis that a decrease in gait speed may predict cognitive decline and dementia risk [28]. Major gait variables such as cadence and joint angles (hip, knee) showed statistically significant differences between the healthy control group and each dementia group, consistent with findings in existing literature. However, for certain variables, the difference between the mild dementia and moderate dementia groups was not statistically significant, suggesting that the boundary between intermediate stages may be relatively unclear.

The ML method (SVM) demonstrated greater effectiveness in classifying individuals with dementia than conventional statistical methods such as ANOVA or logistic regression. This suggests that AI-based approaches are more effective in learning and classifying complex gait patterns. Logistic regression analysis identified knee joint angle and hip joint angle as important explanatory variables in modeling the gait of individuals with dementia. The model’s explanatory power was 57.5%, and its classification accuracy was 77.9%. These results indicate that joint movement during gait is closely associated with cognitive decline, consistent with findings from previous studies. However, traditional statistical models have limitations in capturing nonlinear relationships and multivariate interactions. In contrast, the SVM classifier trained with feature optimization via the Relief algorithm achieved a classification accuracy of 86.3% between individuals with dementia and healthy controls. This represents an 8.4% improvement over logistic regression and illustrates the strength of nonlinear models in classifying complex multidimensional gait data. Machine learning-based analysis offers advantages in modeling high-dimensional variables, detecting atypical patterns, and supporting efficient clinical screening. The SVM classifier used in this study showed comparable performance (86.3%) to the result reported by Vichianin et al., who achieved 85% accuracy in dementia diagnosis using a combination of MRI brain studies and neuropsychological measures [29]. Similarly, the higher classification accuracy of SVM (86.3%) compared to logistic regression (77.9%) aligns with the findings of Mathotaarachchi et al. [30], further supporting the effectiveness of ML in analyzing complex biometric data. Our SVM-based approach with ReliefF feature selection achieved an accuracy of 86.3%, clearly outperforming previous sensor-based studies in this domain. For example, Zhong et al. (2021) reported 75.2% accuracy in MCI detection using traditional statistical analysis of knee kinematic parameters [31], whereas our multiparametric approach leveraging nonlinear feature interactions yielded an 11.1% improvement. Likewise, Bovonsunthonchai et al. (2022) found that spatiotemporal parameters alone provided limited discriminative ability between MCI and cognitively intact groups [32], while our combined spatiotemporal–kinematic feature set successfully classified across the entire cognitive spectrum. A key advancement of our method lies in the application of the ReliefF algorithm to automatically identify optimal feature combinations (velocity, knee joint angle, gait cycle time, hip joint angle) from high-dimensional sensor data, rather than relying on predetermined single parameters.

In terms of dementia severity classification, an SVM classifier was applied using the extracted gait indices as input variables. The classification achieved an accuracy of 86.3%, confirming that dementia severity was correctly predicted in most cases. The model also showed a PPV of 90.6% and a sensitivity of 85.3%, indicating strong performance in accurately identifying relevant cases. These findings suggest that gait-based biomarkers possess high discriminatory power for dementia classification. In particular, the high PPV reflects the model’s reliability in detecting true cases with minimal false positives, while the high sensitivity supports its potential use in clinical screening by reducing false negatives. This combination of performance metrics highlights the value of ML algorithms, particularly SVMs, in the early diagnosis and severity classification of dementia.

A limitation of this study is the absence of direct comparative analysis with other advanced machine learning algorithms, such as Random Forest, XGBoost, or neural networks. Future work should conduct systematic comparisons across multiple algorithms using identical datasets and validation procedures to determine the most effective method for dementia severity classification based on gait analysis data. Such comparative investigations would not only guide optimal algorithm selection but may also reveal ensemble strategies capable of further enhancing classification performance. Our study adheres to established regulatory frameworks for Software as Medical Device (SaMD) clinical validation, which emphasize thorough validation of specific algorithmic implementations through three key pillars: valid clinical association, analytical validation, and clinical validation [33]. Systematic evidence from Al-Hammadi et al.’s review of 40 dementia detection studies highlights SVM as the most widely used machine learning method in this domain, with our achieved accuracy of 86.3% aligning with established performance benchmarks [19]. The sample size of our study (*n* = 139) enables statistically robust single-algorithm validation in accordance with the SSAML methodology [34], while avoiding the statistical pitfalls of inadequately powered multi-algorithm comparisons that often yield wide confidence intervals and potentially misleading performance claims [35]. This focused approach delivers reliable performance estimates that are critical for clinical translation. In terms of limitations, the number of participants was limited, indicating the need for larger-scale studies. Although gait speed can influence gait analysis results, this study allowed participants to walk at their preferred speed to reduce fall risk and support a stable psychological state. Additionally, some gait data were collected in a laboratory environment that may differ from real-world clinical settings. Ensuring that data is collected in consistent gait environments is likely important for data quality. Our laboratory-based data collection approach represents a critical first step in establishing the relationship between gait parameters and dementia severity, consistent with established technical validation frameworks [36]. Conducting experiments under controlled laboratory conditions allows us to isolate specific gait parameters from environmental confounders, which is essential for establishing baseline relationships before real-world validation [37,38]. Moving forward, future research should adopt a staged validation strategy, transitioning from controlled laboratory settings to real-world environments. Such progression will be essential to evaluate the influence of environmental factors and to determine the true clinical utility of our AI-based classification system. Finally, while the Relief algorithm was used to select key gait variables, other potentially important variables may exist, and future studies should explore a broader range of features. Although participants with clearly diagnosed conditions such as fractures or heart disease were excluded, it was not possible to fully control for potential confounding factors that may influence walking ability, including sarcopenia, joint stiffness, or subclinical cardiovascular impairments, which are prevalent among older adults. Furthermore, the use of medications affecting motor or cognitive function may have introduced subtle influences on gait performance. Future studies should implement more comprehensive screening protocols to account for these confounders and incorporate objective assessments of muscle strength, joint flexibility, and cardiovascular fitness to better isolate the specific effects of cognitive decline on gait. A key limitation of this study is the absence of statistical adjustment for demographic confounding factors, particularly age and BMI differences between groups. We did not perform analysis of covariance (ANCOVA) to control for these variables, and therefore cannot completely distinguish whether the observed differences in gait patterns are attributable to cognitive decline or to other factors such as age-related musculoskeletal changes and anthropometric variation [39]. The 12-year age gap between healthy controls and moderate dementia groups likely contributed to the observed joint angle differences since aging independently reduces hip and knee range of motion and alters biomechanical compensation strategies [40]. Future research should address this limitation by implementing statistical covariate adjustment and age-matched designs to more precisely isolate the effects of cognitive decline on gait parameters.

In terms of clinical and research significance, this study demonstrated that dementia severity can be classified more precisely through gait analysis. In particular, it holds considerable medical and technological relevance by showing that AI can complement the limitations of existing statistical techniques. In the future, applying more diverse AI algorithms (e.g., Random Forest, XGBoost) or combining them with multimodal data (e.g., brain imaging, cognitive tests) may further enhance prediction accuracy and clinical applicability. It is also necessary to conduct multi-center studies that collect data from various sites to evaluate the generalizability of the model. In addition, longitudinal studies that track gait changes over time and analyze their correlation with dementia progression are required. This study represents an internal validation conducted under controlled conditions, constituting an essential first phase in clinical ML development [41]. Our validation approach follows established SSAML methodology, employing appropriate cross-validation techniques to generate reliable performance estimates [34]. However, to establish generalizability and clinical applicability, external validation on independent datasets from diverse clinical sites and populations is indispensable. Future work should prioritize multi-center validation studies to rigorously evaluate model performance across diverse patient groups and clinical settings.

The knee and hip joint angle changes identified in this study were consistent with findings showing altered knee kinematics in older adults with mild cognitive impairment during level walking [31]. As dementia severity increases, functional flexibility in the knee joint declines, and gait patterns deteriorate, confirming an interaction. The neurobiological mechanisms underlying the observed knee and hip joint angle changes are consistent with well-established pathophysiological processes in cognitive decline. Tau pathology accumulation in motor control regions, particularly in Braak stages I–IV, has been directly linked to altered gait kinematics in MCI populations [42]. The increased knee extension angle and reduced hip flexion angle identified in the moderate dementia group align with compensatory biomechanical adaptations aimed at maintaining postural stability as executive function deteriorates. These joint angle modifications represent early motor manifestations of disrupted prefrontal–motor cortex connectivity, which is critical for complex movement planning and execution. Importantly, our machine learning approach captures these subtle neuromechanical signatures before they become clinically apparent, providing objective biomarkers of cognitive–motor integration decline. Early exercise interventions are expected to positively influence knee function. During the early and late stages of gait, the knee reaches maximum extension and maximum flexion, while the range of motion tends to narrow during the mid-phase. As a result, the sagittal plane range of motion graph for the knee joint typically shows two peaks, and the overall range may gradually decrease throughout the gait cycle. These findings can serve as valuable clinical information for improving gait characteristics in individuals with dementia.

In conclusion, this study demonstrates that AI-based gait analysis provides a promising and objective assessment approach for dementia severity classification, outperforming traditional statistical methods. By capturing subtle gait patterns and complex feature interactions indicative of cognitive decline, the proposed machine learning framework achieved higher classification accuracy (86.3%) than conventional logistic regression approaches (77.9%). The clinical significance lies in the development of noninvasive, cost-effective, and efficient approaches that can serve as initial screening tools enabling early detection of indicators of cognitive decline that may be overlooked in standard clinical evaluations [43]. Moreover, it aligns with the emerging concept of digital biomarkers, enabling continuous monitoring capabilities and individualized assessments that may be particularly valuable in preclinical dementia stages [44]. While the technical feasibility has been established, future work should emphasize large-scale validation studies and integration into existing clinical workflows to facilitate translation into practical clinical tools.

## 5. Conclusions

In gait analysis, where multiple variables and complex interactions exist, conventional statistical approaches may be limited in handling high-dimensional data and identifying subtle patterns among multiple parameters. In particular, ML techniques, including dimensionality reduction methods such as PCA and gradient-based feature selection, can effectively identify key gait characteristics relevant to dementia severity classification. This study demonstrates that machine learning-based approaches can efficiently manage complex gait datasets and complement traditional statistical analyses by revealing underlying patterns that are difficult to detect using conventional methods alone. These results highlight the potential of AI-based tools to advance gait analysis in dementia research, particularly in studies involving large or multidimensional datasets where conventional statistical approaches may encounter analytical challenges. This valuable contribution to the field is particularly noteworthy considering the limited number of clinical studies that have applied AI for dementia assessment. The findings of this study may serve as important reference data for developing clinical strategies to improve gait characteristics in individuals with dementia. The study also suggests that AI technology can serve as a tool for screening and assessing dementia severity in the elderly population and may form the foundation for developing a quantitative diagnostic support system for clinical application.

## Figures and Tables

**Figure 1 sensors-25-06083-f001:**
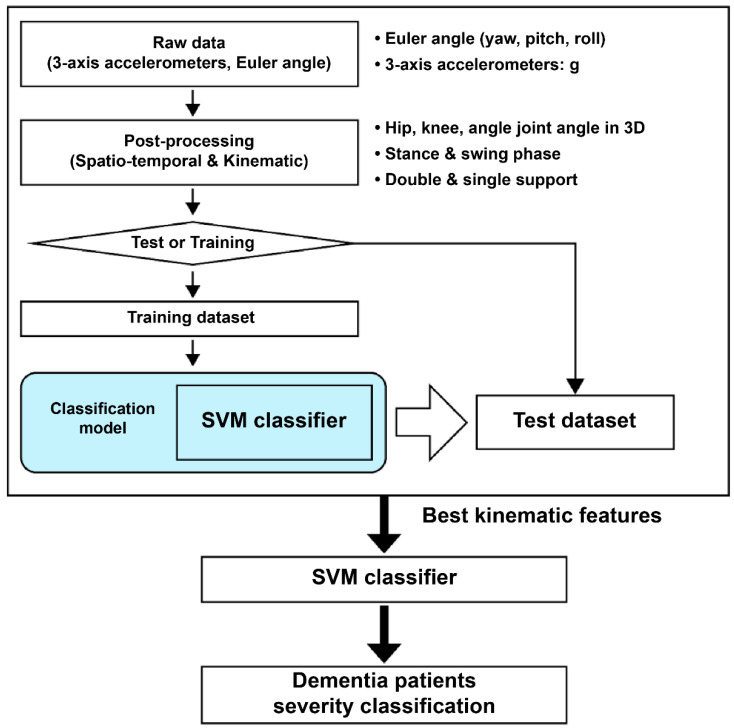
Flowchart illustrating the feature selection process applied to the SVM algorithm to improve the classification performance of dementia severity. SVM: support vector machine.

**Figure 2 sensors-25-06083-f002:**
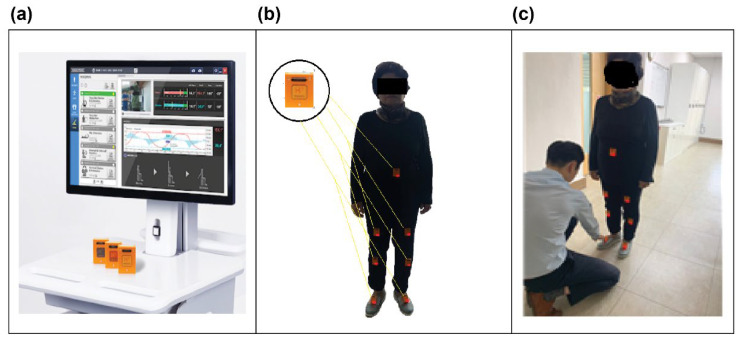
(**a**) Gait analysis system hardware, which includes a PC and wireless IMU sensors (Human Track, R. Biotech Co., Ltd., Seoul, Republic of Korea). Key specifications: IMU model: MPU-9250 9-axis sensor; Sampling frequency: 100 Hz; Data transmission: Bluetooth 4.0; (**b**) IMU sensor attachment locations: lower abdomen (pelvis), bilateral thighs, bilateral tibias, and bilateral feet, totaling 7 sensor units; (**c**) Preparation before subject gait analysis showing sensor calibration procedures.

**Figure 3 sensors-25-06083-f003:**
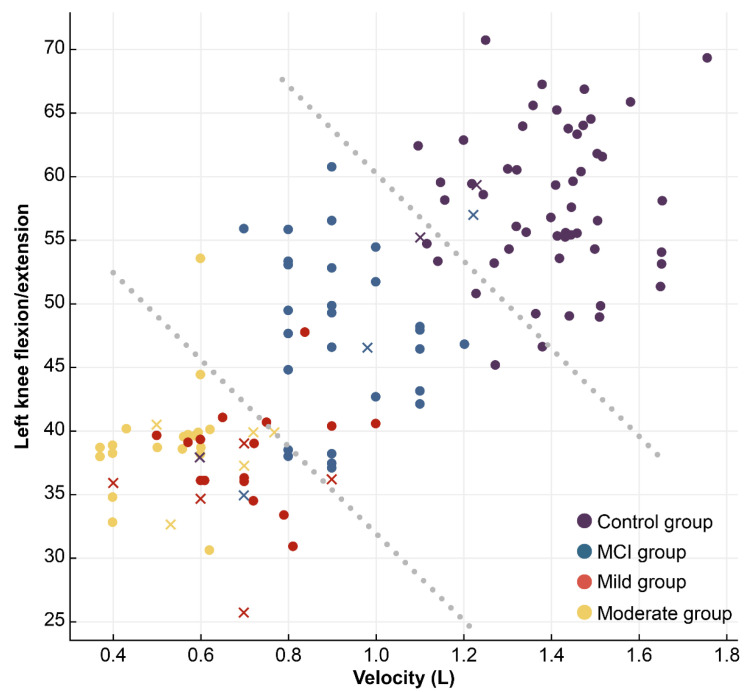
Confusion matrix representation of dementia severity classification results using SVM classifier.

**Figure 4 sensors-25-06083-f004:**
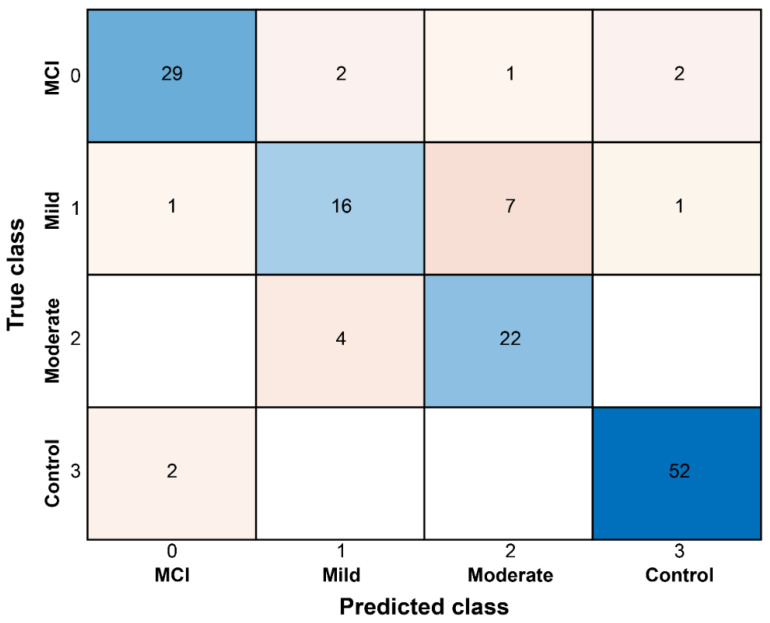
Correct predictions are highlighted in blue in the 4 × 4 confusion matrix.

**Figure 5 sensors-25-06083-f005:**
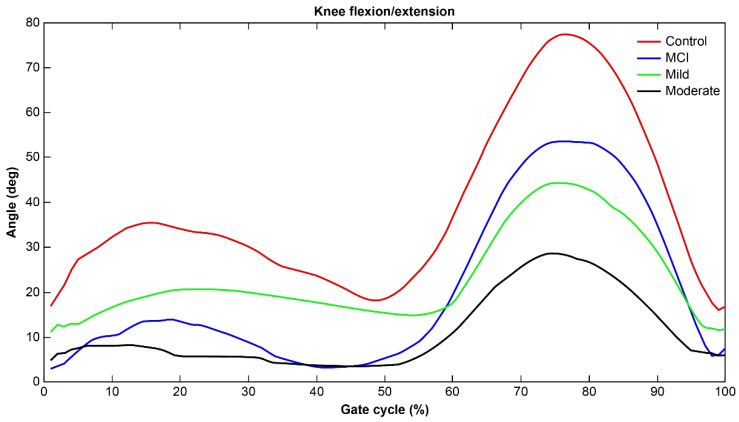
Differences in the graph according to the severity of dementia in the sagittal plane of the knee joint for one representative subject from each group.

**Table 1 sensors-25-06083-t001:** Demographic and Clinical Characteristics of the Study Groups.

Group Number		1	2	3	4	
Group	Total	Healthy Control	MCI	Mild Dementia	Moderate Dementia	*p*-Value
	*n* = 139	*n* = 54	*n* = 34	*n* = 25	*n* = 26	
Age (years)	75.0 (72.0–83.0)	74.0 (72.0–75.0)	75.5 (72.0–82.0)	84.0 (77.0–86.0)	86.0 (76.0–89.0)	<0.0001 *†: b,c,d,e
Gender (male)	49 (35.25)	20 (37.04)	15 (44.12)	4 (16.0)	10 (38.46)	0.13
Gender (female)	90 (64.75)	34 (62.96)	19 (55.88)	21 (84.0)	16 (61.54)	
Height (cm)	157.0 (150.0–165.0)	158.5 (153.0–165.0)	159.0 (152.0–165.0)	150.0 (147.0–158.0)	155.0 (147.0–170.0)	0.0035 *†: b,d
Weight (kg), Mean ± SD	58.7 ± 10.2	60.6 ± 9.8	62.6 ± 9.8	54.3 ± 7.8	54.1 ± 10.8	0.0006 *†: e
BMI (kg/m^2^)	23.1 (21.7– 25.5)	23.2 (22.1–24.8)	24.1 (22.7–26.0)	22.8 (21.5–26.0)	21.7 (19.0–23.8)	0.0087 *†: e

Group Classification Criteria: Participants were classified into four groups based on a comprehensive evaluation using standardized cognitive assessments, including the MMSE and CDR. The classification was as follows: Healthy Control (CDR = 0), MCI (CDR = 0.5), Mild Dementia (CDR = 1.0), and Moderate Dementia (CDR = 2.0). Abbreviations: *n* = number of participants; M = male; F = female; BMI = body mass index; N.S. = not significant. Variables verified for normality are presented as mean ± SD; others are presented as median (Q1–Q3). Statistical Notes: * *p* < 0.0083 indicates a statistically significant difference after Bonferroni correction for multiple comparisons. †: b: Healthy Control vs. Mild Dementia (1 vs. 3); c: Healthy Control vs. Moderate Dementia (1 vs. 4); d: MCI vs. Mild Dementia (2 vs. 3); e: MCI vs. Moderate Dementia (2 vs. 4).

**Table 2 sensors-25-06083-t002:** Kruskal–Wallis test: Difference between each group.

Parameter	Total (*n* = 139)	Healthy Control (*n* = 54)	MCI (*n* = 34)	Mild Dementia (*n* = 25)	Moderate Dementia (*n* = 26)	*p*-Value
Gait cycle time (sec)	1.3 (1.1–1.5)	1.1 (1.1–1.1)	1.3 (1.2–1.3)	1.5 (1.3–1.5)	1.6 (1.5–1.7)	<0.0001 *†: a,b,c,d,e,f
Stance phase (%)	62.3 (60.2–64.3)	61.8 (60.8–62.9)	63.7 (60.9–65.7)	60.6 (58.0–63.9)	63.9 (61.4–67.5)	0.0166
Swing phase (%)	37.7 (35.7–39.7)	38.2 (37.1–39.2)	36.6 (34.3–39.2)	39.5 (36.2–42.0)	36.1 (32.5–38.6)	0.0166
Velocity (m/s)	1.0 (0.7–1.2)	1.2 (1.2–1.3)	1.0 (0.9–1.1)	0.7 (0.6–0.8)	0.6 (0.4–0.6)	<0.0001 *†: a,b,c,d,e,f
Cadence (steps/min) (mean ± SD)	100.1 ± 12.2	110.4 ± 6.1	99.2 ± 4.6	89.8 ± 7.9	83.8 ± 9.1	<0.0001 *†: a,b,c,d,e
Initial double support (%), (mean ± SD)	12.5 ± 3.6	12.1 ± 2.3	12.6 ± 4.1	11.5 ± 3.8	13.7 ± 4.2	0.0576
Single support (%)	37.2 (35.3–39.1)	37.5 (36.3–38.7)	36.6 (35.3–40.3)	37.7 (35.8–40.0)	36.0 (33.6–38.1)	0.2343
Terminal double support (%)	12.5 (10.2–14.4)	12.3 (10.7–13.5)	13.0 (9.3–14.6)	11.6 (9.0–13.6)	13.5 (11.9–16.9)	0.0471
Hip joint angle (deg)	42.0 (33.4–47.1)	47.8 (45.0–51.1)	42.4 (38.1–44.8)	33.3 (27.3–36.7)	28.6 (26.0–35.2)	<0.0001 *†: a,b,c,d,e
Knee joint angle (deg)	48.0 (39.4–54.2)	53.9 (51.5–56.6)	48.1 (44.6–54.4)	39.4 (35.6–40.3)	38.9 (35.4–39.9)	<0.0001 *†: a,b,c,d,e
Ankle joint angle (deg) (mean ± SD)	27.0 ± 7.0	32.1 ± 3.7	23.8 ± 4.2	23.9 ± 5.6	23.5 ± 9.3	<0.0001 *†: a,b,c

Variables verified for normality are presented as mean ± SD; others are presented as median (Q1–Q3). Statistical Notes: * *p* < 0.0083 indicates a statistically significant difference after Bonferroni correction for multiple comparisons. †: a: Healthy Control vs. MCI (1 vs. 2); b: Healthy Control vs. Mild Dementia (1 vs. 3); c: Healthy Control vs. Moderate Dementia (1 vs. 4); d: MCI vs. Mild Dementia (2 vs. 3); e: MCI vs. Moderate Dementia (2 vs. 4); f: Mild Dementia vs. Moderate Dementia (3 vs. 4).

**Table 3 sensors-25-06083-t003:** Healthy control vs. Dementia groups.

	Univariable Analysis		Multivariable Analysis			
	OR (95% CI)	*p*	OR (95% CI)	*p*	R-Square	Max-Rescaled R-Square
Knee joint angle (deg)	1.269 (1.172–1.374)	<0.0001	1.098 (0.905–1.332)	0.0076	0.5745	0.7787
Hip joint angle (deg)	1.510 (1.304–1.749)	<0.0001	1.244 (1.030–1.503)	0.023		

Abbreviations: OR = odds ratio; CI = confidence interval; R-Square and Max-rescaled R-Square indicate model explanatory power.

**Table 4 sensors-25-06083-t004:** Accuracy of dementia severity classification using SVM.

Classifier	Accuracy (%)	Precision (PPV) (%)	Recall (Sensitivity) (%)	Specificity (%)	F1-Score (%)	AUC	NPV (%)
SVM	86.33 (%)	90.62 (%)	85.30 (%)	91.47 (%)	87.89 (%)	0.924	88.75 (%)

## Data Availability

The original contributions presented in this study are included in the article. Further inquiries can be directed to the corresponding author.

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
