# Peer review of "AI-Based Severity Classification of Dementia Using Gait Analysis"

_sensors, 2025, doi:10.3390/s25196083_

Round 1
Reviewer 1 Report
Comments and Suggestions for Authors
This paper discusses the classification method of dementia severity based on AI and gait analysis, combining machine learning methods with traditional statistical methods, which is clinically significant but insufficiently innovative, and suggests the following:
- Figure 1 can be supplemented with more detailed annotations, such as IMU model and sampling frequency.
- Although the accuracy of SVM classifier (86.3%) is better than that of logistic regression (77.9%), this method is not compared with other machine learning algorithms, and it is recommended to add comparison experiments.
- The SVM performance indicators in Table 4 only list accuracy, PPV, and sensitivity, and do not provide complete evaluation indicators such as specificity or F1 score, so it is recommended to supplement them.
The overall English expression is good, but there are some sentences with awkward word order and unidiomatic expressions.
Reviewer 2 Report
Comments and Suggestions for Authors
Overall
This research addresses a very contemporary and important topic: classifying the severity of dementia by combining gait data from IMU sensors with machine learning. We highly commend its focus and results. Furthermore, we would like to offer some comments and suggestions for further strengthening the paper's logical structure and arguments, leading to a higher-quality academic contribution.
(1) Emphasis on the academic contribution and novelty of this study
Points to be noted
Text (ambiguity of contribution):
`This is considered a valuable contribution, particularly given the current lack of clinical studies applying AI in the context of dementia.`
Text (lack of explanation of novelty):
`This study demonstrates that machine learning-based classification can effectively identify subtle patterns and variable interactions that are difficult to detect using traditional statistical techniques, through dimensionality reduction and advanced feature selection.` (lines 308-311)
A) Clarification of the contribution to the journal to which this paper is being submitted (Sensors):
The current description only states that this study's contribution is "the application of AI to clinical dementia research," which would be appropriate in a general AI or healthcare journal. If this paper is submitted to Sensors, the focus should be more clearly on how sensor technology has brought about a breakthrough in dementia research.
[Example of suggested improvement]
Please add the following points to the `Introduction` and `Conclusion`.
"Traditional dementia assessment relies on subjective and fragmented information such as interviews and observations. In contrast, the IMU sensors used in this study enable objective and continuous capture of three-dimensional movement information in millisecond increments. This high-resolution time-series data captures subtle variations in walking patterns that traditional statistical methods have overlooked, enabling the construction of highly accurate machine learning models..."
B) Emphasizing Novelty in Data Processing
Statistical processing using "average values" can overlook important information such as individual variations and outliers. The core of this research lies not in simple average value comparisons, but in feature selection using the ReliefF algorithm and nonlinear classification using SVM. This process of "extracting essential features from high-dimensional data and recognizing complex patterns" is the novelty of this research.
[Example Improvement Proposal]
Please specify the `Discussion` section as follows: "The unique feature of this study is that rather than simply comparing statistical averages, the ReliefF algorithm was used to automatically select the features (e.g., speed, knee joint angle) that most contribute to severity classification from high-dimensional data consisting of 11 gait parameters, and then an SVM was used to learn the nonlinear relationships between the parameters. ...Please explain this section more precisely.
(2) Analysis of Features and Clarification of Findings from Sensor Data
Points of Remark
Main Text: `Table 1. Demographic and Clinical Characteristics of the Study Groups.` (line 113) and `Table 2. Kruskal-Wallis Test: Difference Between Each Group`
Points of Remark and Improvement
Currently, the basic characteristics of the subject groups (Table 1) and the inter-group comparison of each gait parameter (Table 2) are presented separately. However, the logical development of the study would be clearer if we first discussed the statistical characteristics of each group regarding the important features (e.g., speed, knee and hip joint angles) "discovered" by the machine learning model.
[Improvement Suggestion]
We propose restructuring the `Results` section as follows to present the findings from the sensor data in stages.
At the beginning of `3.1 Kruskal-Wallis Test`, we state, "Focusing on walking speed, knee joint angle, and hip joint angle, which were later found to be important in machine learning analysis, clear statistical differences were observed between the healthy and dementia groups." Based on the results of Table 2, we first describe how these key features change depending on the severity of dementia.
Then, in section `3.3 Classification of Dementia Severity Using AI`, we conclude, "The greatest discovery from the sensor data analysis in this study is that not only the differences in these individual parameters but also the combination of these features enables us to classify severity with a high accuracy of 86.3%, exceeding conventional statistical models."
This flow clearly conveys how the value of the analysis progresses from "differences in individual sensor indicators" to "pattern recognition through combinations of sensor indicators."
(3) Differentiate your research from existing studies and clarify the pioneering nature of your sensor use.
`Recent research on gait has utilized wearable devices to identify repetitive patterns within the gait cycle...`
`The IMU-based gait analysis system is a practical tool for detecting abnormal gait patterns...`
In the `Introduction`, you mention the existence of research using wearable sensors, but there is insufficient discussion of how this research will bring about breakthroughs compared to existing dementia research (especially research that does not use sensors). To emphasize the pioneering nature of your sensor use, you need to specifically contrast the limitations of conventional methods with how sensors overcome them.
Please add the following contrasting discussion to the fourth paragraph of your `Introduction`.
"Traditional gait assessment in dementia research has relied heavily on measuring walking speed using a stopwatch, the Timed Up and Go (TUG) test, or visual observation by clinicians. [Quote] While these methods are simple, they often result in fragmented assessments, are subjective to the evaluator's subjectivity, and are difficult to capture dynamic, qualitative changes during the gait cycle, such as subtle asymmetries and disruptions in joint coordination. The IMU sensor used in this study overcomes these limitations..."
(4) Clarification of Sensor Data Acquisition Conditions
Points of Recommendation
Text: `2.2. Gait Analysis` `All participants in this study completed eight gait trials along an 8-m gait course. `The experiment was conducted in a quiet environment...` `All participants performed the procedures under identical conditions throughout the analysis.`
Points of Recommendation/Improvement
Sensor data is susceptible to environmental noise, so strict uniformity of experimental conditions is crucial to ensuring the reliability of research. While statements such as "quiet environment" and "under identical conditions" are correct, more detailed information is needed to ensure reproducibility in a scientific paper.
Please add the following specific experimental conditions to the `2.2. Gait Analysis` section:
"The 8-m walkway was placed on a level, non-slip floor (e.g., linoleum)."
Environment: "The room temperature was maintained at 24±2°C, the humidity at 50±10%, and the illuminance was standardized at 500 lux."
"Subjects wore comfortable clothing and comfortable sneakers while walking."
"Subjects were instructed to 'walk at a comfortable pace as usual,' and to focus their gaze approximately 5 m ahead."
"A one-minute seated rest was provided between each trial to minimize the effects of fatigue."
Adding these details will significantly improve the rigor and reproducibility of the study.
(5) Discussion of confounding factors, such as comorbid conditions in the subjects.
``Exclusion criteria included a diagnosis of orthopedic conditions... circulatory system conditions such as heart disease...''
This study attempts to control for comorbid conditions that may affect walking by setting exclusion criteria, which is commendable. However, elderly people with dementia potentially have a variety of physical factors that may affect walking, such as muscle weakness, joint contractures, and mild cardiopulmonary dysfunction, even if they do not reach the point of diagnosis. These factors cannot be completely excluded, and they may affect the interpretation of the results.
[Improvement Suggestion]
The existence of these potential confounding factors should be mentioned in the ``Limitations`` section of the ``Discussion` section and honestly discussed as a limitation of the study.
"Fifth, although this study excluded clearly diagnosed diseases such as fractures and heart disease, it was not possible to fully control for potential confounding factors that may affect walking ability, such as age-related muscle weakness (sarcopenia), limited joint range of motion, or undiagnosed mild cardiovascular disease, which are common among older adults. ..."
---
(6) Discussion on Data Preprocessing
Points to Note
Main Text: Throughout this paper, the specific process for converting raw sensor data into analyzable features such as "gait cycle time" and "knee joint angle" is not described.
Points to Note/Improvements
This is a significant shortcoming in the reproducibility of this study. The core of IMU sensor data analysis is how to detect meaningful gait events (e.g., heel strike, toe-off) from noisy acceleration and angular velocity signals and calculate kinematic parameters.
Please add a new section such as "2.3 Data Preprocessing and Feature Extraction" and describe the following processing flow in detail:
"A fourth-order Butterworth low-pass filter with a cutoff frequency of XX Hz was applied to the acceleration and angular velocity data obtained from the IMU sensor to remove high-frequency noise."
"The start point (Initial Contact) and end point (Terminal Contact) of the gait cycle were automatically detected from the filtered foot angular velocity data using the XX algorithm [quote]."
"Based on the detected gait cycle, spatiotemporal parameters such as walking time, walking speed, and cadence were calculated."
4. Calculating joint angles: "The orientation (cross-sectional area) of the sensor for each body part was calculated."
Round 2
Reviewer 1 Report
Comments and Suggestions for Authors
1.Using only SVM limits the comparative assessment against other ML algorithms (e.g., Random Forest, XGBoost, neural networks).
2.The model was validated only internally; external validation on independent datasets is needed to ensure clinical applicability.
3.Data were collected in a lab setting; validation in real-world clinical environments is recommended.
Reviewer 2 Report
Comments and Suggestions for Authors
Overall
(1) Impact of the study and comparison with previous research (Reviewer points 1, 3)
> Author's response 1: "We have revised the Introduction and Conclusion sections to better highlight the sensor technology breakthrough."
> Author's response 3: "We have incorporated a contrasting discussion into the Introduction to better emphasize how sensors overcome the limitations of conventional methods."
> Introduction: "Conversely, the IMU sensors used in this study enable objective and continuous capture of three-dimensional movement information... thereby supporting the development of highly accurate machine learning models." (p.2)
> Discussion: "A key strength of this study lies in moving beyond simple comparisons of statistical averages. Instead, the ReliefF algorithm was applied to automatically identify the features... This approach of 'extracting essential features from high-dimensional data and recognizing complex patterns' represents the central novelty of this research." (p.12)
In response to the reviewer's comments, the authors added a contrasting statement in the introduction between the limitations of traditional gait analysis methods, such as stopwatches and TUG tests (e.g., subjectivity and fragmented assessment), and the technological breakthroughs offered by IMU sensors (objectivity, continuity, and high resolution). This makes the significance of using sensor technology itself clearer than before. The authors also praised the positioning of the combined ReliefF and SVM approach in the discussion section as the "central novelty of this study." However, the reviewers were looking for more than a simple explanation of the general conflicts between "sensors vs. traditional methods" or "machine learning vs. traditional statistics." More importantly, they wanted to understand why this research is novel compared to previous studies using other sensors or machine learning. Even in the revised paper, there is a lack of specific comparisons with other similar studies (e.g., [15, 22]), leaving the impression that the contribution is limited to "using AI for gait analysis."
In the `Discussion` section, you should discuss the advantages and novelty of your research, citing specific methods and results, such as: "In the study by [reference number], SVM classification was performed using only walking speed, but the accuracy rate was only 75%. In contrast, in this study, we used ReliefF to extract important features such as knee joint angle from multidimensional gait parameters, achieving a higher accuracy rate of 86.3%. This demonstrates the importance of capturing nonlinear combinations of multidimensional features rather than a single parameter, and this is our contribution to previous research."
As the reviewer pointed out, there is a lack of discussion of why a particular method is superior or its mechanism. Wouldn't adding a neuroscientific and biomechanical perspective in the `Discussion` section about "why knee and hip angles are associated with cognitive decline" be a scientific contribution that goes beyond simply describing the phenomenon?
(2) Lack of discussion regarding subject homogeneity (Reviewer points 2, 5)
> Author's response 5: "We have added a comprehensive discussion of potential confounding factors to the Limitations section."
> Discussion/Limitations: "Although participants with clearly diagnosed conditions such as fractures or heart disease were excluded, it was not possible to fully control for potential confounding factors, including sarcopenia, joint stiffness, or subclinical cardiovascular impairments... Future studies should implement more comprehensive screening protocols..." (p.14)
The report merely lists potential confounding factors and lacks discussion of the more fundamental issue of the homogeneity and variability of basic demographic information (e.g., age, gender, BMI) of the subject population. For example, Table 1 shows that the median age differs significantly between the healthy control group (74.0 years) and the moderate dementia group (86.0 years), a difference that is statistically significant (p < .0001). It is essential to discuss the extent to which this heterogeneity between the groups influenced the differences in gait patterns.
In the `Results` section, please mention the demographic differences between the groups shown in Table 1 and discuss their clinical implications. For example, you should state, "The moderate dementia group was significantly older than the healthy control group, and this age difference itself may have contributed to the decline in walking ability."
In the `Discussion` section, you should also add a more self-critical observation, such as, "In this study, we did not statistically adjust for the influence of confounding factors such as age and BMI (e.g., analysis of covariance). Therefore, we cannot completely distinguish whether the observed differences in gait patterns are purely due to cognitive decline or to other factors such as age. This is a limitation of this study."
(3) Paper Structure and Arrangement of Figures and Tables
Figure 2 (SVM flowchart) is located in Section 2.4, which explains the methodology. Its content illustrates the overall flow of the study, namely, "feature selection" and "classification." Moving this to the end of the introduction or the beginning of the methodology would make it easier for readers to grasp an overview of the entire analysis process.
Tables 2, 3, and 4 and Figures 3, 4, and 5 are summarized in the `Results` section, but the flow of discussion is somewhat complicated: "statistical tests → logistic regression → SVM." Wouldn't it be easier to understand if a clear roadmap was provided at the beginning of the section, explaining what question each analysis is trying to answer?
